# A novel home-use culture mechanism for identifying microbial load in urine samples

**Monika Singh, Riya Sahai, Siddharth Pattnaik** *

Samplytics Technologies Pvt. Ltd., Koramangala, Bangalore, India

* siddharth@inito.com

**Data Availability Statement:** All relevant data are within the paper and its Supporting Information files.

**Funding:** The authors received no specific funding for this work.

## Abstract

Diagnosing a urinary tract infection (UTI) is typically a clinical procedure involving multiple steps. The need to perform a test depends on the presence of relevant symptoms. Given the current pandemic situation, visiting a clinic may not be a preferable choice for many users. Many vulnerable groups of patients, namely, males with certain predispositions and pregnant women, may not present with symptoms of UTI and may go undiagnosed, which could give rise to a more complicated situation. So far, microbial cultures have been used as the gold standard for the diagnosis of an infection. However, performing microbial cultures currently requires trained professionals and laboratory grade equipment. Therefore, there is a need for a home-use culture kit that can serve this purpose. To our knowledge, no such kit exists. Here, we present a feasibility study of an affordable and easy-to-use home-based setup for quantifying bacterial load in a urine. We believe that such a system can be used by people at home to monitor recurrent UTI infections and also by physicians to remotely monitor and prescribe narrow-spectrum antibiotics for more effective treatment.

## Introduction

Microbial cell culture is a versatile tool widely used to detect, identify, and quantify pathogenic loads for research and diagnostic purposes. Inoculating clinical specimens into a pathogen-specific culture media aids the visualisation of these pathogens to assess the microbial origin and severity of the disease.

Standard culture-based diagnostic methods can detect microbes in urine samples, stools, the genital tract, deep throat, and skin. An example of microbial infection is urinary tract infection, which affects the ureters, urethra, urinary bladder, or, in extreme cases, kidneys. Several risk factors are associated with uncomplicated UTIs. Women are known to be more prone to UTI due to an anatomical bias which makes them more susceptible [1]. The risk of occurrence of UTI also increases as a woman ages [2]. Additionally, pregnant women are usually at a much higher risk of contracting UTI due to shoter urethra and excretion of high amounts of proteins and sugars [3–6]. Apart from women, those with diabetes, elderly men, patients on immunosuppressants and infants are also prone to recurrent UTI [7,8]. UTI is acquired by up to 50–60% of individuals at least once in their lifetime [9]. If undiagnosed, UTI can complicate to a renal infection and therefore individuals at risk need to undergo regular screening to

**Competing interests:** The authors have declared that no competing interests exist.

avoid further complications or progression to complicated UTIs. Furthermore, in people with recurrent UTIs, antimicrobial resistance may be a major concern due to exposure to broad spectrum antibiotics time and again. Therefore a precise diagnosis of the species is also required.

A common method of diagnosing UTI is using urinalysis dipsticks [10,11]. Predictive urinalysis is used most commonly to decide the treatment for uncomplicated UTIs. However, these tests provide inaccurate results nearly 33% of the time, and their sensitivity ranges from 23–95.6%, depending on the type of patient [12,13]. A significant limitation of this method is its inability to determine the causative agent, which may lead to the prescription of an incorrect treatment strategy that aggravates the development of multidrug-resistant bacterial species, further increasing the burden of antimicrobial resistance [14]. Urine cultures, in contrast, help identify the exact causative agent. Since UTI symptoms also coincide with the early symptoms of many sexually transmitted diseases and vaginal infections, urine cultures are necessary to differentiate UTIs from other related disorders [15]. For these reasons, urine culture is considered the 'gold standard' for diagnosing UTIs [16,17]. However, the traditional way of performing a microbial culture requires skilled technicians and considerable time (around 48 hours) for results to reach the patient.

Certain commercial products have tried to alleviate this cumbersome traditional process for instance dip-slides [18]. However, dip-slides incompletely tackle this problem since the samples need to be shipped back to the laboratory for colony counting and pathogen identification. In addition, these tests are designed for a midstream sample and hence require volumes in the range of 50–60 mL. Therefore, there is a need for an easy-to-use home based culturing mechanism which is user-friendly as well as accurate enough to be used by physicians.

This study presents a novel home-based method for culturing and detecting microbes. In this study, we focused on microbes causing UTIs, specifically *Escherichia Coli* to demonstrate the dynamic nature of this system regarding microbes and culture media. Our primary goal was to design a system derived from the dipstick format and integrate it with a culture medium, which can also be extended to other cultures and infections. We first found that microbes did not need to be spread on top in order to be cultured and could grow at the bottom of the agar plates. Such an observation has not been shown before. We also found that the setup conceived through this work could be used to deliver urine samples to the culture media without the need of pipetting it on the media and spreading it making it a more user-friendly and less technically challenging workflow. In addition, we found that the system could be used to culture bacteria at room temperature which makes it a great option for home use.

## Materials and methods

### Bacterial cell culture

The *E. coli* strain K-12 was used in this study. A single colony was inoculated to obtain uniform growth. To isolate pink colonies, 10 μL of the culture was spread on a MacConkey agar plate. For all experiments, pink colonies were selected from the plate and grown in a 600 mL Luria Bertani (LB) medium. The cultures were incubated in a shaker incubator maintained at 37°C with rotation at 180 rpm. The culture was monitored every 2 h until an optical density (OD) of approximately 0.65–0.75 was achieved. Optical density was measured in triplicates. To compare the OD values, sterile blank LB solution was incubated under similar conditions. This was used as blank for all measurements. For all experiments, a blank plate and a plate with the blank solution were incubated along with the different experimental setups to ensure the sterility of the incubator and culture media.

## Culture media preparation

For all experiments with *E. Coli*, three different agar media were used: Luria Bertani (LB) agar (M1151, HiMedia, USA), MacConkey agar (M7408, Millipore, USA), and HiChrome Agar (M1353R, HiMedia, USA). For preparing the LB agar. For the preparation of MacConkey Agar, 30 gm of the agar was mixed with 600 mL distilled water. For the preparation of HiChrome Agar, 34.08 gm was mixed in 600 mL distilled water. All the agar media and liquid cultures were autoclaved at 121˚C and 15 psi for 30 min.

## Membrane selection

For all experiments with membranes, Whatman Grade 470 was used to deliver the sample to agar media. This was selected based on the pore size of the membrane. Before integrating the membrane into the experiment it was washed thoroughly with 70% ethanol, followed by washing with distilled water.

## Image capturing and analysis

All cultured plates were imaged using an in-built phone camera. A customised LED-mounted translucent light diffuser setup (designed for home use) was used to capture images of the cultured plates (Fig 1B). Images were captured using iPhone 12. The images were captured at an exposure of 1/875s and ISO 32. Colonies were counted manually using OpenCFU 3.0 [19]. OpenCFU is an open source software for counting cell colonies or any other circular object. No post-capture changes were applied to the images before processing with OpenCFUs. The images were processed in a jpeg format. A circular region of interest (ROI) was defined in the software with a "regular" threshold since the colonies had a darker appearance (red) compared to the medium (white). OpenCFU returns results in the form of $x_{/y}$ where x is the number of valid colonies and y is the total number of circular objects captured. For consistency, only the x value from all the images were taken ahead for statistical analysis. The size of colonies for detection was set to 4 since in our case the colonies formed were smaller and setting a higher threshold led to missing more positive colonies. The radius was auto-set by OpenCFU.

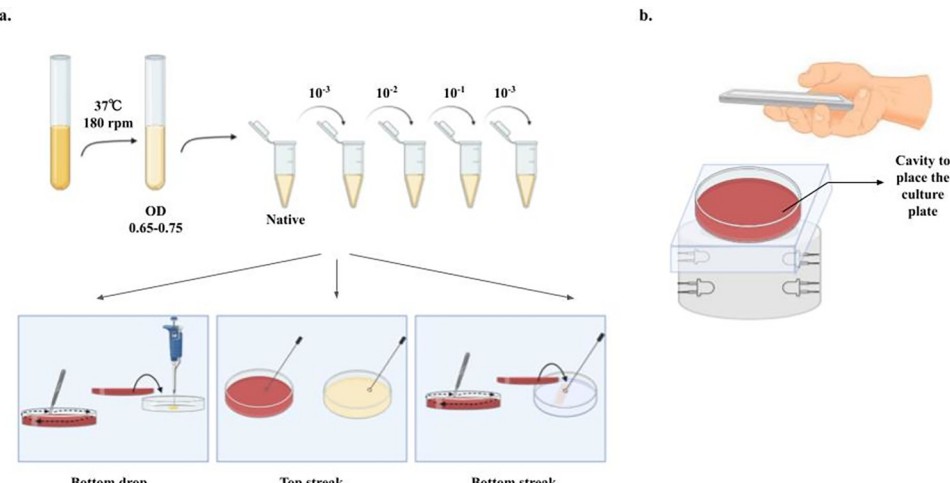

**Fig 1. Summary of experimental design and imaging setup.** Illustration of experimental design to generate serial dilutions and of different application methods (a). Schematic of the imaging set up used for capturing images of the cultured plates using a mobile phone (b).

### Statistical analysis

All statistical analyses were performed using GraphPad Prism version 9. For all groups of experiments, an unpaired, non-parametric t-test (Kolmogorov–Smirnov test) was performed to compare the cumulative distributions and determine the p-value. Results were considered statistically significant at a p-value of 0.1 and a 90% statistical significance.

## Results

### Microbial growth is unaffected by the location of the culture application

Since the design of our system was dependent on the delivery of the sample from the bottom to avoid any contamination from air or external sources, we aimed to prove that culturing bacteria below an agar medium can be used to estimate the bacterial load. In addition, we checked whether sensitivity was maintained in terms of colony density between the two methods. To achieve this, we diluted the cultured bacterial media and compared the top streaking method with the bottom application. The bottom application was performed in two ways: by streaking and pouring the culture media. In both cases, agar was placed after the culture was applied to the plate (Fig 1A).

We streaked the culture on top of the Luria Bertani (LB) agar plates as a control to ascertain that there was no contamination. For the initial run, MacConkey agar plates were used because they are commonly used to diagnose UTIs [20]. Interestingly, we found that the dilutions were visibly distinct from one another in all different setups (Fig 2A).

For quantification, we selected dilutions where distinct colonies could be counted: $1:10^5$, $1:10^6$, and $1:10^9$. Interestingly, we found that the number of colonies was also similar in all methods of sample application (Fig 2B), indicating that culture grown below an agar medium has similar sensitivity and performance as the canonical culture application. In addition, manual counting and OpenCFU-based quantification yielded similar results (Fig 2B).

We then examined whether our observations were media-specific. To assess this, we chose HiChrome chromogenic agar media, a commonly used growth medium for diagnosing the causative agent of UTI. The experimental setup for this growth medium was similar to that shown in Fig 1A. With the chromogenic agar media, we replicated the observations with MacConkey agar (S1A and S1B Fig), indicating that the bottom culture application method is also true for different types of agar media.

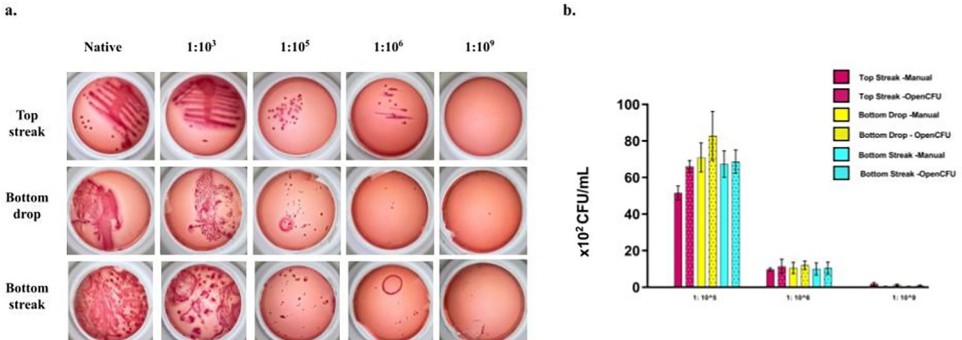

**Fig 2. Growth of E.coli on MacConkey agar delivered as per experimental design.** Growth of cultured E.coli on MacConkey agar applied using various methods i.e. top streak, bottom streak and bottom drop and over serial dilutions (a). Comparison of CFU/ml obtained from different application methods and using different quantification methods. CFU/ml was significantly different (p<0.1) between all dilutions for both the quantification methods as well as for all methods of application. CFU/ml calculated manually and by OpenCFU were not significantly different (p>0.1).

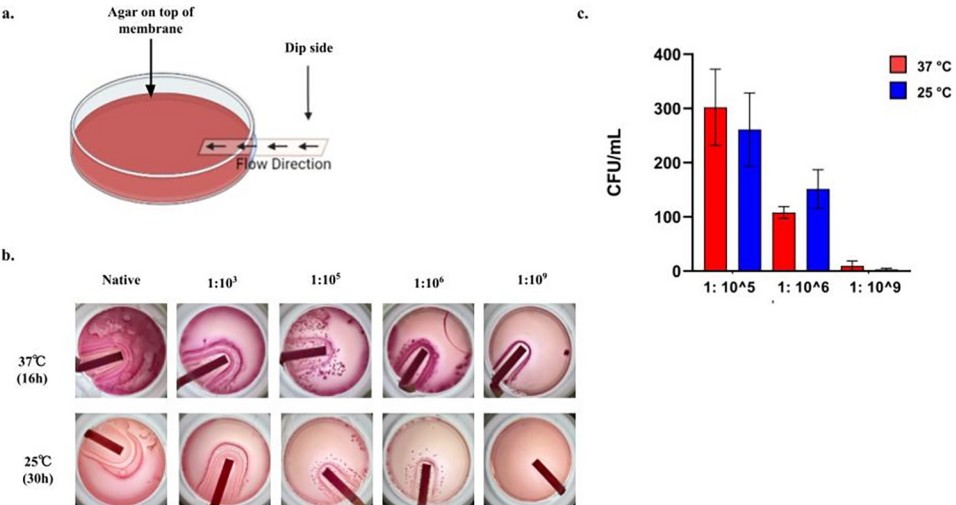

**Fig 3. Membrane based culture delivery and corresponding growth of E. coli at different temperatures in MacConkey agar.** Schematic of the membrane-based delivery apparatus (a). Growth of cultured E. coli on MacConkey agar at 37˚C for 16 hours and at 25˚C for 30 hours with the sample delivered using the membrane (b). Comparison of CFU obtained manually for both temperature conditions. CFU was significantly different (p<0.1) between all dilutions in which colonies could be counted for both the conditions except between 1:$10^5$ and 1:$10^6$ at 25˚C (p = 0.5). CFU calculated for both conditions were not significantly different in any dilution (p>0.1).

## Membrane-mediated bottom delivery performs similar to direct application

In a home-use setup, pipetting a fixed volume of liquid onto or below the agar medium is usually not feasible. Therefore, we investigated whether a membrane-mediated delivery could solve this issue. For this purpose, we developed the system shown in Fig 3A, where the membrane jutting out of the Petri dish was used to deliver the sample to the agar media.

The protruding end of the membrane was dipped in bacteria-containing media for 15 s. We examined the setup using different agar media to establish the robustness of this method. Similar culture dilutions were used in the previous experiments. We observed a visible gradient in the colony count for each dilution in all the different agar media used (Figs 3B and S2A).

Since in this setup the native culture, 1:$10^3$, and 1:$10^5$ dilutions developed dense colonies, we could not consider these concentrations for further quantification. The computed number of colonies was greater than in the previous experimental designs discussed since a greater culture volume was being delivered to the sample pad. Hence, we demonstrated that membrane-mediated sample delivery works equally well with the top or bottom application of the culture without affecting the sensitivity of the entire system.

## Membrane-delivered microbial samples can be cultured at room temperature

To facilitate the use of the system at home, cultures needed to be grown at room temperature. Therefore, we investigated the performance of the membrane-based delivery mechanism at room temperature. The aim was to determine the earliest time point at which distinct colonies were visible. We found that colonies were visible within 36 h of the sample application. Interestingly, we discovered that at 36 h, the colonies obtained were similar in number to those cultured at 37˚C for 16 h, indicating that 36 h is probably long enough for any bacterial load to manifest. However, we found that at lower temperatures, the difference between dilutions

$1:10^5$ and $1:10^6$ (although existent) was not as statistically significant as the difference obtained by growing the culture at 37˚C. This could imply that this system is slightly less sensitive than growth at higher temperatures and may have to be tweaked accordingly. It is also possible that a larger number of replicates may overcome this variation.

## Discussion

In this study, we present a novel system to visualise and quantify the bacterial load in urine samples at home to detect and identify UTI-related pathogens. To the best of our knowledge, such a system does not exist for the diagnosis and management of UTIs.

Although the results presented through this study validate the accuracy of CFU measurement using the home-use system, there are some limitations which need to be kept in view. The system presented consistent results in spiked urine and cultured bacteria. However, it needs to be validated with clinical samples, which will be the scope of a future study. It is possible that the sensitivity observed may be different, in which case the composition of the culture media may need to be adjusted. Our results indicated that the sensitivity of the assay varied slightly when cultured at different temperatures namely 25˚C and 37˚C. Therefore the CFU/ml to determine a cut-off for positive or negative diagnosis may vary from the laboratory standard and will have to be established as part of a clinical trial. Additionally, we assume that the room temperature is 25˚C. While this is the commonly accepted standard, practically homes may have varying temperatures based on the external temperature or usage of temperature modulating mechanisms. Such considerations would have to be taken into account when designing the product for usage at home and the results may vary slightly based on the actual room temperature. Another issue that needs to be addressed is the hygienic disposal of culture plates in home-use scenarios. However, this can be solved operationally by providing a bag for safe disposal or a tube of bleach that can be applied once the results have been secured.

As compared to existing diagnostic techniques for UTI, we believe that the system proposed through this work provides a more specific, sensitive and easy to use design which can be used at home. The design of the proposed diagnostic system is such that it does not need trained professionals unlike canonical culturing protocols. In addition, compared to the time taken by currently available culturing methods, we believe that the proposed mechanism can be performed in a much shorter time. While it may be argued that urine dipsticks work in the shortest possible time, the curtailing of reading time comes at the cost of specificity to species detection and sensitivity of the colony forming units per mL. This may cause urine dipsticks to return more false negative results than can be accepted as has been reported by previous works as well. Table 1 shows a comparison of urine dipstick, canonical culture techniques and the novel technique proposed based on multiple factors.

We foresee many potential applications for such home-use systems. For instance, certain groups of patients who are more vulnerable to UTI or are at risk of asymptomatic UTI can

**Table 1. Comparison between the novel home based culture method, traditional microbial cultures and urine dipstick.**

|  | Novel home use culture method | Traditional microbial cultures | Urine dipstick |
|---|---|---|---|
| **Type of detection** | Culture-based | Culture-based | Colorimetric |
| **Time to result** | 24h | 48h | 3–5 min |
| **Specificity to bacterial species** | Yes | Yes | No |
| **Can differentiate bacterial species** | Depends on agar used | Depends on agar used | No |
| **Requires specialised equipment** | No | Yes | No |
| **Requires trained personnel** | No | Yes | No |
| **Sensitivity** | <10 CFU/mL | <10 CFU/mL | $10^4$–$10^5$ CFU/mL |

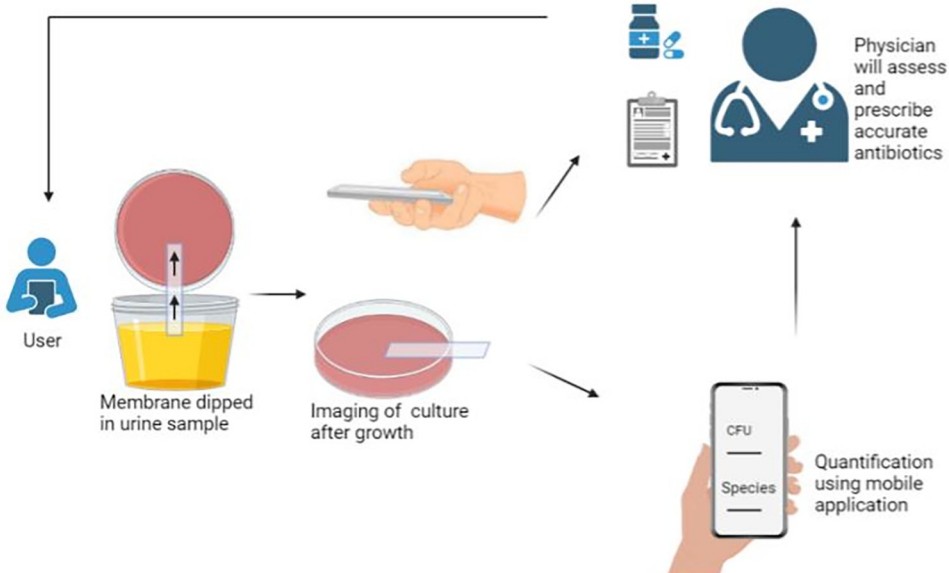

**Fig 4. Flowchart of putative home usage of the membrane-based delivery system.** A representative flow chart of the interaction between the user and the physician with the membrane-based delivery system aiding the diagnosis and intervention process.

keep these at home and monitor the presence or absence of infection regularly. The interface can be made following a Stackelberg game approach with a monitoring on the participation of the users [20]. In the case of an infection, the results can be shared with the respective physician. The entire sharing platform can be secured using the BSIF system [21]. As an extension, the results from this system can be used by physicians to remotely identify the causative agent, assess the microbial load, and suggest the most effective treatment [22,23] (Fig 4).

Antibiotic resistance in uropathogens has been widely studied and is a concern [24–26]. Since our system can help in identifying the exact species that cause infection, it can be used to prescribe species-specific antibiotics. This would help reduce the burden of antibiotic resistance, preventing the development of multidrug-resistant pathogens, which would be especially useful in patients with recurrent UTIs.

As a primary goal, we have also demonstrated that bottom culture is feasible and the number of colonies formed is comparable to that of a standard top streak. Recently, many tests have been based on analysing dried urine samples, where the samples can be transported on a filter card [27]. One of our future aims is to study the efficacy of this system with dried urine samples on a filter card. A comparative analysis of the microbial load in liquid and dried urine samples may be worthwhile to make this system more beneficial. Future studies will also focus on a mobile application-based system that can be used to quantify the microbial load and determine the severity of infections.

## Conclusion

In conclusion, we present a microbial load monitoring setup which can be used to extrapolate the CFU/ml in a given urine sample without the hassle of using canonical culture methodology or apparatus. We show that the system can be reliably used to measure colony count even at room temperature and therefore is suitable to be at home. Additionally, because the system does not require samples to be applied directly, we believe this is a more hygienic mechanism since it avoids spillage or direct contact with the user. We believe that such a system can be

used to identify the bacterial species and prescribe narrow spectrum antibiotics in cases of recurrent infections. Future research work will focus on using this system in a clinical setup where two groups, one using the proposed system and one getting diagnosed by a clinic, will be compared. It would be interesting to see if the two methods can be used interchangeably. We will also be conducting additional clinical trials to evaluate the effectiveness of this home-use culture mechanism in prescribing antibiotics. This would be helpful in establishing the efficacy of this system with respect to prevention of antimicrobial resistance. Parallely, we would also be interested to investigate the probable use of this system to diagnose fungal infections.

## Supporting information

**S1 Fig. Growth of E.coli on HiChrome UTI agar delivered as per experimental design.** (DOCX)

**S2 Fig. Membrane based culture delivery and corresponding growth of E.coli at different temperatures in HiChrome UTI agar.** (DOCX)

## Author Contributions

**Conceptualization:** Siddharth Pattnaik.

**Data curation:** Riya Sahai.

**Formal analysis:** Siddharth Pattnaik.

**Investigation:** Monika Singh, Riya Sahai, Siddharth Pattnaik.

**Methodology:** Monika Singh, Siddharth Pattnaik.

**Project administration:** Monika Singh, Siddharth Pattnaik.

**Resources:** Siddharth Pattnaik.

**Supervision:** Siddharth Pattnaik.

**Validation:** Riya Sahai, Siddharth Pattnaik.

**Visualization:** Riya Sahai.

**Writing – original draft:** Siddharth Pattnaik.

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
