## [Decision Letter · Decision Letter 0]

6 Nov 2022

PONE-D-22-22330A novel home-use culture mechanism for identifying microbial load in urine samplesPLOS ONE

Dear Dr. Pattnaik,

Thank you for submitting your manuscript to PLOS ONE. After careful consideration, we feel that it has merit but does not fully meet PLOS ONE’s publication criteria as it currently stands. Therefore, we invite you to submit a revised version of the manuscript that addresses the points raised during the review process.

ACADEMIC EDITOR: The authors have to address the comments from the reviewers carefully. Please submit your revised manuscript by Dec 21 2022 11:59PM. If you will need more time than this to complete your revisions, please reply to this message or contact the journal office at plosone@plos.org. Please include the following items when submitting your revised manuscript:A rebuttal letter that responds to each point raised by the academic editor and reviewer(s). You should upload this letter as a separate file labeled 'Response to Reviewers'.A marked-up copy of your manuscript that highlights changes made to the original version. You should upload this as a separate file labeled 'Revised Manuscript with Track Changes'.An unmarked version of your revised paper without tracked changes. You should upload this as a separate file labeled 'Manuscript'.

We look forward to receiving your revised manuscript.

Kind regards,

Thippa Reddy Gadekallu

Academic Editor

PLOS ONE

Journal Requirements:

3. Please ensure that you have specified (1) whether consent was informed and (2) what type you obtained (for instance, written or verbal, and if verbal, how it was documented and witnessed). If your study included minors, state whether you obtained consent from parents or guardians. If the need for consent was waived by the ethics committee, please include this information.

Reviewers' comments:

Reviewer's Responses to Questions

**Comments to the Author**

1. Is the manuscript technically sound, and do the data support the conclusions?

Reviewer #1: Yes

Reviewer #2: Yes

2. Has the statistical analysis been performed appropriately and rigorously? 

Reviewer #1: Yes

Reviewer #2: Yes

3. Have the authors made all data underlying the findings in their manuscript fully available?

Reviewer #1: Yes

Reviewer #2: Yes

4. Is the manuscript presented in an intelligible fashion and written in standard English?

Reviewer #1: Yes

Reviewer #2: Yes

5. Review Comments to the Author

Reviewer #1: In this work, the author proposed a novel home-use culture mechanism for identifying microbial load in urine samples. The comments are listed as follows:

1. The contribution should be summarized in several sentences in brief, which can help the reader better understand the significance of this paper.

2. The existing works should be summarized and presented with the time or logic order rather than the direct illustration, the reviewer thinks that one table for the current efforts comparison may be better.

3. What are the shortcomings or research gaps for the current related work? I think the authors should highlight them at the end of each subsection of the related work section.

4. Where is the source of the adopted data? Please add some explanations.

5. What are the detailed parameter settings for the adopted neural networks? The authors should add more details to the manuscript.

6. To broaden the scope of this paper, the authors should refer to some papers. For example, AoI optimization in the UAV-aided traffic monitoring network under attack: A Stackelberg game viewpoint; BSIF: Blockchain-based Secure, Interactive, and Fair Mobile Crowdsensing;

Mixed Game-based AoI Optimization for Combating COVID-19 with AI Bots; Data Freshness Optimization Under CAA in the UAV-Aided MECN: A Potential Game Perspective.

7. This paper needs to be carefully proofed and polished before being accepted.

Reviewer #2: The work is interesting and scientifically good. However, the structure of the article should be redesigned

1)The abstract should focus on problem and how it is solved.

2)The introduction part requires related and motivation.

3)The methodology must be explained in detail and the author should specify how their approach is different.

4)The results should be compared with existing works.

5)The threats to validity must also be present in detail

6)The conclusion and the future work should be presented in detail.

Deep neural networks to predict diabetic retinopathy , be used to modify the structure of the paper.

6. PLOS authors have the option to publish the peer review history of their article (what does this mean?). If published, this will include your full peer review and any attached files.

Reviewer #1: No

Reviewer #2: No

---

## [Author Response · Author response to Decision Letter 0]

20 Dec 2022

Dear Professor Gadekallu,

Thank you so much for giving us an opportunity to make revision to our manuscript. We are really grateful to all the reviewers and editors for their comments and support. We have now addressed the major comments raised by the reviewers and have edited our manuscript accordingly. The changes have been tracked in the revision that we are submitting. In addition, we are also submitting a clean copy of the manuscript with the changes made. We hope that the changes are satisfactory to the editors. Below is a detailed description of each comment and our response to that:

Reviewer #1: In this work, the author proposed a novel home-use culture mechanism for identifying microbial load in urine samples. The comments are listed as follows:

1. The contribution should be summarized in several sentences in brief, which can help the reader better understand the significance of this paper. - We are grateful to the reviewer for this comment. We have changed the last paragraph of our introduction in order to summarize the findings of this paper. We believe that this fits well with the flow of the introduction.

2. The existing works should be summarized and presented with the time or logic order rather than the direct illustration, the reviewer thinks that one table for the current efforts comparison may be better. 

3. What are the shortcomings or research gaps for the current related work? I think the authors should highlight them at the end of each subsection of the related work section. - We would like to thank the reviewer for this comment. We have added the shortcomings of the existing works and have highlighted them in the introduction for reference. In addition, we have also added a detailed description of the research gaps of our work in the discussion section.

4. Where is the source of the adopted data? Please add some explanations. - We are grateful to the reviewer for pointing this out. We have added the source of the data used in the materials and methods and have also explained the parameters. 

5. What are the detailed parameter settings for the adopted neural networks? The authors should add more details to the manuscript. - Again, we are thankful for this comment. We agree that the parameter settings were not clearly explained and we have added the setting details in the materials and methods.

6. To broaden the scope of this paper, the authors should refer to some papers. For example, AoI optimization in the UAV-aided traffic monitoring network under attack: A Stackelberg game viewpoint; BSIF: Blockchain-based Secure, Interactive, and Fair Mobile Crowdsensing;

Mixed Game-based AoI Optimization for Combating COVID-19 with AI Bots; Data Freshness Optimization Under CAA in the UAV-Aided MECN: A Potential Game Perspective. - We would like to thank the reviewer for this comment. While we have used the papers to structure the manuscript now, we feel that the papers cannot be referred to since the scope of these works are very different from our current research.

7. This paper needs to be carefully proofed and polished before being accepted.

Reviewer #2: The work is interesting and scientifically good. However, the structure of the article should be redesigned

1)The abstract should focus on the problem and how it is solved. - We would like to thank the reviewer for this comment. We agree that the abstract in our initial submission was limited. However, we have made it more elaborate to highlight the problem and how our proposed method alleviates the issue.

2)The introduction part requires related and motivation. - We are thankful to the reviewer for suggesting this. We have edited the paragraphs to be more connected and we hope that the different motivations are conveyed clearly by each paragraph. 

3)The methodology must be explained in detail and the author should specify how their approach is different. - We have added more details to the methodology and we hope the details suffice to understand the methods used in quantification of the paper.

4)The results should be compared with existing works. - We would like to thank the reviewer for this comment. The results are quite unique in their own self. However, as suggested by reviewer 1, we have added a comparison table in the discussion section comparing existing works and our work. We have also added a detailed explanation of the same.

5)The threats to validity must also be present in detail - We understand the reviewer’s concerns. We have added the threats to validity in the discussion section in more detail and have also suggested what further studies need to be conducted to invalidate those threats to the work here.

6)The conclusion and the future work should be presented in detail. - Once again, we are grateful to the reviewer for pointing this out. We have edited the conclusion section to bring forth the future prospects of this system as well as plausible extension of the work described.

Deep neural networks to predict diabetic retinopathy , be used to modify the structure of the paper.

Once again, we are very grateful to you for your kind consideration and to the reviewers for diligently pointing out important gaps in our manuscript. We hope you find everything in order.

Yours sincerely,

Siddharth Pattnaik

Corresponding author.

---

## [Decision Letter · Decision Letter 1]

3 Feb 2023

PONE-D-22-22330R1A novel home-use culture mechanism for identifying microbial load in urine samplesPLOS ONE

Dear Dr. Pattnaik,

Thank you for submitting your manuscript to PLOS ONE. After careful consideration, we feel that it has merit but does not fully meet PLOS ONE’s publication criteria as it currently stands. Therefore, we invite you to submit a revised version of the manuscript that addresses the points raised during the review process.

ACADEMIC EDITOR: Authors have to carefully address all the concerns raised by the reviewers.   Please submit your revised manuscript by Feb 22 2023 11:59 PM. If you will need more time than this to complete your revisions, please reply to this message or contact the journal office at plosone@plos.org. Please include the following items when submitting your revised manuscript:A rebuttal letter that responds to each point raised by the academic editor and reviewer(s). You should upload this letter as a separate file labeled 'Response to Reviewers'.A marked-up copy of your manuscript that highlights changes made to the original version. You should upload this as a separate file labeled 'Revised Manuscript with Track Changes'.An unmarked version of your revised paper without tracked changes. You should upload this as a separate file labeled 'Manuscript'.

We look forward to receiving your revised manuscript.

Kind regards,

Kwame Kumi Asare, Ph.D

Academic Editor

PLOS ONE

Reviewers' comments:

Reviewer's Responses to Questions

**Comments to the Author**

1. If the authors have adequately addressed your comments raised in a previous round of review and you feel that this manuscript is now acceptable for publication, you may indicate that here to bypass the “Comments to the Author” section, enter your conflict of interest statement in the “Confidential to Editor” section, and submit your "Accept" recommendation.

Reviewer #1: (No Response)

Reviewer #2: All comments have been addressed

2. Is the manuscript technically sound, and do the data support the conclusions?

Reviewer #1: Yes

Reviewer #2: Yes

3. Has the statistical analysis been performed appropriately and rigorously? 

Reviewer #1: No

Reviewer #2: Yes

4. Have the authors made all data underlying the findings in their manuscript fully available?

Reviewer #1: Yes

Reviewer #2: Yes

5. Is the manuscript presented in an intelligible fashion and written in standard English?

Reviewer #1: Yes

Reviewer #2: Yes

6. Review Comments to the Author

Reviewer #1: In this work, the author proposed a novel home-use culture mechanism for identifying microbial load in urine samples. The comments are listed as follows:

1. The contribution should be summarized in several sentences in brief, which can help the reader better understand the significance of this paper.

2. The existing works should be summarized and presented with the time or logic order rather than the direct illustration, the reviewer thinks that one table for the current efforts comparison may be better.

3. What are the shortcomings or research gaps for the current related work? I think the authors should highlight them at the end of each subsection of the related work section.

4. Where is the source of the adopted data? Please add some explanations.

5. What are the detailed parameter settings for the adopted neural networks? The authors should add more details to the manuscript.

6. To broaden the scope of this paper, the authors should refer to some papers. For example, AoI optimization in the UAV-aided traffic monitoring network under attack: A Stackelberg game viewpoint; BSIF: Blockchain-based Secure, Interactive, and Fair Mobile Crowdsensing;

Mixed Game-based AoI Optimization for Combating COVID-19 with AI Bots; Data Freshness Optimization Under CAA in the UAV-Aided MECN: A Potential Game Perspective.

7. This paper needs to be carefully proofed and polished before being accepted.

Reviewer #2: The work is interesting and scientifically good. However, the structure of the article should be redesigned

1)The abstract should focus on problem and how it is solved.

2)The introduction part requires related and motivation.

3)The methodology must be explained in detail and the author should specify how their approach is different.

4)The results should be compared with existing works.

5)The threats to validity must also be present in detail

6)The conclusion and the future work should be presented in detail.

Deep neural networks to predict diabetic retinopathy , be used to modify the structure of the paper.

7. PLOS authors have the option to publish the peer review history of their article (what does this mean?). If published, this will include your full peer review and any attached files.

Reviewer #1: No

Reviewer #2: No

---

## [Author Response · Author response to Decision Letter 1]

22 Apr 2023

Dear Professor Asare,

Thank you so much for giving us an opportunity to make revision to our manuscript. We are really grateful to all the reviewers and editors for their comments and support. We have now addressed the major comments raised by the reviewers and have edited our manuscript accordingly. The changes have been tracked in the revision that we are submitting. In addition, we are also submitting a clean copy of the manuscript with the changes made. We hope that the changes are satisfactory to the editors. Below is a detailed description of each comment and our response to that:

Reviewer #1: The authors have not well addressed all my comments, some important references are missing. The authors should refer to the previous comments for paper improvement again.

We are grateful to the reviewer for their comments. We have now added the references as suggested by the reviewer in their previous comments. We hope that the reviewer finds the references in place.

Reviewer #2: The authors have addressed all the suggestions provided in the previous review. I recommend the article to be accepted

We are really grateful to the reviewer for their previous comments and for recommending our article for publication.

Once again, we are very grateful to you for your kind consideration and to the reviewers for diligently pointing out important gaps in our manuscript. We hope you find everything in order.

Yours sincerely,

Siddharth Pattnaik

Corresponding author.

---

## [Decision Letter · Decision Letter 2]

4 May 2023

A novel home-use culture mechanism for identifying microbial load in urine samples

PONE-D-22-22330R2

Dear Dr. Siddharth Pattnaik,

We’re pleased to inform you that your manuscript has been judged scientifically suitable for publication and will be formally accepted for publication once it meets all outstanding technical requirements.

Kind regards,

Kwame Kumi Asare, Ph.D

Academic Editor

PLOS ONE

Additional Editor Comments (optional):

Reviewers' comments:

Reviewer's Responses to Questions

**Comments to the Author**

1. If the authors have adequately addressed your comments raised in a previous round of review and you feel that this manuscript is now acceptable for publication, you may indicate that here to bypass the “Comments to the Author” section, enter your conflict of interest statement in the “Confidential to Editor” section, and submit your "Accept" recommendation.

Reviewer #1: All comments have been addressed

2. Is the manuscript technically sound, and do the data support the conclusions?

Reviewer #1: No

3. Has the statistical analysis been performed appropriately and rigorously? 

Reviewer #1: No

4. Have the authors made all data underlying the findings in their manuscript fully available?

Reviewer #1: No

5. Is the manuscript presented in an intelligible fashion and written in standard English?

Reviewer #1: Yes

6. Review Comments to the Author

Reviewer #1: all comments have been well addressed, so it can be accepted. all comments have been well addressed, so it can be accepted.all comments have been well addressed, so it can be accepted.

7. PLOS authors have the option to publish the peer review history of their article (what does this mean?). If published, this will include your full peer review and any attached files.

Reviewer #1: No

---

## [Editor Report · Acceptance letter]

9 May 2023

PONE-D-22-22330R2 

A novel home-use culture mechanism for identifying microbial load in urine samples 

Dear Dr. Pattnaik:

I'm pleased to inform you that your manuscript has been deemed suitable for publication in PLOS ONE. Congratulations! Your manuscript is now with our production department. 

Kind regards, 

on behalf of

Dr. Kwame Kumi Asare 

Academic Editor

PLOS ONE